# Trunk and Upper Body Fatigue Adversely Affect Running Economy: A Three-Armed Randomized Controlled Crossover Pilot Trial

**DOI:** 10.3390/sports7080195

**Published:** 2019-08-19

**Authors:** Scott N. Drum, Ludwig Rappelt, Lars Donath

**Affiliations:** 1School of Health & Human Performance, Northern Michigan University, 1401 Presque Isle Ave., Marquette, MI 49855, USA; 2Department of Intervention Research in Exercise Training German Sport University Cologne, 50933 Köln, Germany

**Keywords:** oxygen uptake, running economy, lactate, endurance, fatigue, trunk strength

## Abstract

Trunk muscle fatigue and its negative relationship with running economy (RE) is frequently recognized by practitioners but lacks evidence-based support. Thus, this three-armed randomized controlled crossover pilot trial (RCT) examined the effects of trunk and upper body fatigue protocols on RE, trunk muscle isometric rate of force production, and lactate response in runners. Seven well-trained runners (2 males and 5 females) randomly underwent control (CON), trunk fatigue (TRK), and upper body fatigue (UPR) protocols on three different lab visits. Both workload-matched fatigue protocols—consisting of 24 min of a circuit weight routine—elicited comparable rates of perceived exertion, heart rate responses, and lactate accumulations. As expected, core muscle strength assessed with isometric testing immediately before and after both fatigue protocols, decreased notably. RE (VO_2_/kg bodyweight averaged for 1 min) was determined during a 15 min individual anaerobic threshold (IAT) run at 4, 9 and 14 min. The IAT (13.9 to 15.8 km/h) was determined on lab visit one using an incremental treadmill running protocol to volitional exhaustion. RE differed, although not significantly, between CON and both fatigue protocols by 0.75 (4th min) to 1.5 ml/min/kg (9th and 14th min) bodyweight (Time × Mode Interaction: *p* = 0.2, n_p_^2^ = 0.40) with a moderate to large effect size. Despite no signficance, the largest RE differences were observed between TRK and CON (and underscored by the moderate to large effect size). This preliminary pilot RCT revealed that both UPR and TRK conditions might adversely impact running economy at a high intensity, steady state running pace. Future studies should elucidate if these findings are replicable in large scale trials and, in turn, whether periodized core training can beneficially preserve RE.

## 1. Introduction

Along with established maximal oxygen uptake (VO_2max_) [1] and individual anaerobic threshold (IAT) concepts [2], running economy (RE) assessment gained increasing popularity during the last decade [3] and should play a major role within aerobic capacity testing in endurance sports. RE is considered a multifactorial concept with a multitude of underlying metabolic, respiratory, neuromuscular, and biomechanical characteristics. Numerous RE determinants are trainable (e.g., ventilation, metabolism) and some are of anthropometric nature (e.g., tendon length, calf circumference). RE typically refers to a steady state oxygen consumption at a given submaximal running velocity (within 1 min or over 1 kilometer/mile) and has been measured at different constant paces between 10 and 20 km per hour [3].

Beside core temperature, heart rate, ventilation, lower limb moment arms, and muscle activity patterns during foot striking, muscle properties and fatigue components have also been discussed as modulators of RE [4,5]. Although only a small cross-sectional association between core strength and performance exists [6], the trunk has been frequently considered essential in terms of force transmission and stability, as a “chain is only as strong as its weakest part” [7]. Interestingly, few research groups have investigated the effects of a trunk fatiguing protocol on sports specific endurance performance [8]. To the best of our knowledge, current literature is lacking in the effect of trunk fatigue on RE at IAT in well-trained runners. 

As a case in point, the researchers Prieske, Muehlbauer, and Granacher (2016) [6] recently completed an interesting systematic review and meta-analysis regarding “the role of trunk muscle strength for physical fitness and athletic performance in trained individuals”. Alluded to above, they found that although trunk muscle strength is indeed enhanced by core strength training programming, very little improvement in physical fitness or athletic performance was indicated. Notably, most of the trained or athletic populations included in their review and statistical calculations included team sports (e.g., basketball, football, baseball, volleyball), with only two running investigations and no overreaching discussion about running economy. Furthermore, both running investigations used in the analysis recruited sub-elite or recreational runners. 

Tong and colleagues (2016) [9], whose study was part of the review and meta-analysis [6] mentioned earlier, assessed running economy at onset of blood lactate accumulation as an outcome variable in recreational runners when implementing the use of “functional” inspiratory training coupled with treadmill interval sessions and core muscle exercises over a 6-week (wk) period. Indeed, they found augmented running economy concurrent with improved core and inspiratory muscle function. In contrast, an investigation by Stanton and collaborators (2004) [10], found no improvement in running economy after a 6-week Swiss ball intervention despite a significant improvement in core stability. Moreover, this group utilized young males (i.e., on average 15-years old) participating in baseball and touch football programs. 

Thus, several studies have looked at the effects of core muscle strengthening and stability routines over about a 6-week training cycle related to alterations in running economy. The systematic review and meta-analysis included mostly team sport and non-endurance athletes. No research, that we encountered, used well-trained, competitive runners undergoing a running economy test at a competition level pace. 

Hence, the present randomized controlled crossover pilot study was designed to recruit well-trained distance runners with a competitive history and current/continuous training to investigate whether two different local fatigue protocols (i.e., trunk vs. upper body) would alter RE versus a control condition. Additionally, we sought comparable lactate responses, among other measured stress variables, after the two, 24 min fatigue sessions. It was hypothesized that pre-fatigue, in general, alters RE and trunk fatigue would further encumber RE. This preliminary small sample pilot cross-over trial was primarily conducted to provide data for sample size estimation in future large scaled studies on trunk fatigue and RE.

## 2. Materials and Methods

### 2.1. Study Design and Participants

The present study was conducted as a randomized controlled crossover pilot trial with a repeated measures design. This study was approved by the local ethical committee (Ethical Proposal no. 101/2019). Thereby, large changes of RE upon fatigue would be detectable with 6 subjects assuming pilot study power of 80% with an alpha significance level of *p* < 0.05. On the first lab visit, participants were familiarized with fatigue protocols after undergoing an incremental maximal oxygen uptake (VO_2max_) treadmill test until volitional exhaustion [11]. On the second, third and fourth lab visits, runners performed either a trunk fatigue (TRK), upper body fatigue (UPR) or control (CON) protocol. All testing days were randomly assigned, completed within one month with approximately 1-week rest between lab visits, and at similar times of day. TRK muscle assessments on resistance machines included isometric abdominal flexion, back extension, and abdominal twist (left and right), performed identically and immidately before and after the 24-min fatiguing protocol related to TRK and UPR. RE testing was immediately conducted after the post-isometric strength testing of the TRK muscles, which concluded the 24-min fatigue bout. Lastly, throughout all testing days, participants wore a heart rate transmitter chest strap and receiver/watch (Polar Electro Inc., Bethpage, NY, USA). 

Seven well-trained, competitive runners were enrolled in this study. Their characteristics (mean ± SD) included: sex = 2 males and 5 females; age = 28.2 ± 8.1 years, BMI = 21.9 ± 3.7, running VO_2max_  =  61.3  ±  4.2 mL·min^−1^·kg^−1^, and annual training  =  470  ±  80 hours (hrs) over 72.0 ± 86.5 months and 6.4 ± 0.8 days·week^−1^. Therefore, we assumed this group to be highly trained and stable with regard to running adaptations. All participants were informed about the study and signed a written consent to participate. Athletes refrained from intense training 24 h prior to the testing days and were instructed to continue training, per their coach’s plan, as usual for the month of testing. Notably, from informal questioning, each runner abstained from hard efforts at least 24-h before all lab visits. Lastly, we chose to group males and females together, similar to other researchers observing running economy in a mixed-gender study [12]. 

### 2.2. Testing Procedures

#### 2.2.1. Incremental Exercise Testing 

In order to determine VO_2max_ and the individual anaerobic threshold (IAT), participants performed an incremental exercise test on a running treadmill until voluntary exhaustion on their first lab visit. The initial step was 10 kilometer per hour (km/h) and was increased every 3 min by 2 km/h. Objective exhaustion [11] was verified if the majority of the following 6 exhaustion criteria (4 out of 6) were met: rating of perceived exertion (CR-10 scale) > 8 [13], maximum lactate (lactatepeak, mmol·L^−1^) concentration > 10 mmol·L^−1^, maximum heart rate (heart rate peak, in beats·min^−1^) derived from the empirical formula 208 − (0.7 × age) [14], the respiratory exchange ratio (RER) of carbon dioxide output and oxygen uptake > 1.1, pulmonary ventilation (VE) equivalent for oxygen (VE/VO_2_) > 35, and respiratory frequency (f) > 35 breaths·min^−1^. All subjects fulfilled these prerequisites for objective exhaustion (i.e., at least 4 out of 6 criteria). VO_2max_ values were derived from a breath by breath spirometric system (Zan 600, Zan Messgeräte, Oberthulba, Germany) and capillary blood samples from the earlobe were collected every 3-min for lactate analysis (EBIOplus; EKF Diagnostic Sales, Magdeburg, Germany) while the participant momentarily straddled the treadmill between incremental stages. The respiratory gas exchange instrumentation was calibrated according to the manufacturer’s guidelines with calibration gas. The highest consecutive oxygen uptake values within 30 s at the final step were considered as VO_2max_. IAT determination is discussed later.

#### 2.2.2. Trunk Fatigue Protocol

Familiarization of TRK and UPR protocols took place on lab visit one after the initial VO_2max_ assessment. TRK exercises were performed in a circuit weight training pattern, using multiple sets and reps, standardized by a metronome cadence of 1-s concentric action followed by a 2-sec eccentric action, at a ratio of 45-s lifting:15-s rest = 1 min/set, which elicited 15 reps/set in a speed-controlled manner with 15-s rest between sets and exercises. Notably, the same two researchers supervised and coached participants through the two (i.e., TRK and UPR) fatigue protocols. For TRK, each participant went through the following order of machine exercises and sets: [ab twist left × 3 sets, ab twist right × 3 sets, ab front × 3 sets, back ext × 3 sets] × 2 = 24 min total workout. Average and peak heart rate and perceived effort were recorded in addition to immediate post-exercise lactate via an earlobe lancet stick. Further, immediately prior to and after the 24-min fatigue bout, isometric strength was assessed (Edition-Line, gym80, Gelsenkirchen, Germany) for ab twist left, ab twist right, ab front, and back ext. Participants were similarly instructed and coached throughout the testing to give maximal, isometric action for 3 successive max efforts. The highest of three rates of force development (RFD) measures was used in our final analysis to compare pre to post-RFDs, thereby illustrating the magnitude of fatigue from the resistance protocol. On average, within 5-min post-isometric strength testing, the runner began the 15-min RE test.

#### 2.2.3. UPR Body Fatigue Protocol

UPR was conducted in the exact manner as TRK, including assessment of heart rate, lactate, sets, reps, rest, isometric core RFD assessment at pre- and post-workout (as described prior), and 15-min RE test within approximately 5-min of completing the post-fatigue-workout RFD assessment. The only differences were machines and muscle groups used during the fatigue bout. Thus, during UPR, each participant underwent the following order of machine exercises and sets: seated biceps curl × 3 sets, seated bench press × 3 sets, seated biceps curl × 3 sets, lat-pulldown × 3 sets, seated bench press × 3 sets, seated biceps curl × 3 sets, lat-pulldown × 3 sets, seated bench press × 3 sets. Note, biceps curl and bench press were completed over 3 × 3 sets while lat-pulldown was completed with 2 × 3 sets = 24 min total work. 

#### 2.2.4. Control Condition

To keep in line with TRK and UPR interventions, we had participants during CON visit the circuit weight training lab prior to their 15-min RE bout. Instead of a fatigue protocol, participants engaged in relaxed stretching exercises for 24-min. They also underwent the same maximal, isometric testing (of the core muscles: ab twist right, ab twist left, ab front, back ext) before and after the stretching session and began the RE bout within approximately 5-min of leaving the strength training lab.

#### 2.2.5. Running Economy (RE) Test at Individual Anaerobic Threshold (IAT)

The RE bout occurred three times, approximately one week or greater a part, at approximately 5-min post-intervention (i.e., TRK, UPR, CON). IAT was determined from the initial incremental, running exercise test to volitional maximum oxygen uptake on a motor-driven treadmill. In brief, IAT was determined from plotting (via standardized graph paper) lactate values versus running velocity (pace); whereby a line was drawn tangent to the blood lactate curve to the point where the recovery lactate value was equal to the greatest observed lactate concentration during the test [15]. The same researcher plotted and determined all IAT outcomes. Each runner started the RE test at exactly their predetermined IAT pace while wearing a gas analysis mask and heart rate monitor. Each athlete was encouraged verbally and often by the researchers throughout each test. RPE (Borg, 6-20) was assessed after every 5-min interval. RE data (mL·min^−1^·kg^−1^ oxygen uptake) was averaged every fifth minute (i.e., 4–5, 9–10, 14–15 min) for analysis. Lastly, blood lactate samples (assessed via the earlobe) were taken immediately prior to and after the RE bout. Lactate concentration was analyzed as described earlier using the equipment indicated. 

### 2.3. Statistics

Demographic (provided earlier) and performance data are provided as means with standard deviations (SD). All outcome parameters were initially analyzed using SPSS 13.0 (IBM) (Version 13, IBM, Armonk, NY, USA) for normal distribution (Kolmogorov-Smirnov test) and variance homogeneity (Levene test). Separate 2 (mode: TRK vs. UPR body) × 2 (time: pre vs. post) repeated measures analyses of variances (rANOVA) for crossover trials were calculated to investigate whether differences in fatigue occurred after the TRK and UPR body protocol, respectively. For RE as the primary endpoint, a separate 3 (CON, TRK, UPR) × 3 (5 min, 10 min, 15 min) rANOVA was computed. Significance level for the rANOVA was set a *p* < 0.05 and effect sizes were judged as follows: large effect, ηp² > 0.14, moderate, ηp² > 0.08, small, ηp² < 0.08. In case of significant time × group interactions for the respective parameters, Tukey honest significant difference (HSD) post-hoc tests were additionally performed accompanied by computing standardized mean differences as a measure of pairwise effect size estimation. For pairwise effect size estimation standardized mean differences (SMD) were also computed (SMD, trivial: d < 0.2, small: 0.2 ≤ d < 0.5, moderate: 0.5 ≤ d < 0.8, large d ≥ 0.8) [16].

## 3. Results

### 3.1. Incremental Maximal Test

As described prior, a maximal oxygen uptake, incremental treadmill test was employed during lab visit 1. Table 1 summarizes variables measured during the protocol. 

### 3.2. Fatigue Protocols

Lactate response (mean ± SD) after TRK (2.9 ± 1.9 mmol·L^−1^ ) and UPR (3.5 ± 1.7 mmol·L^−1^) at a similar total external workload did not differ (*p* > 0.05). Data characteristics for the fatigue protocols are given in Table 2. Furthermore, large and significant mode (UPR vs. TRK) × time (pre vs. post) interaction effects were found for three TRK testing conditions (Table 3). Post-hoc tests only revealed statistically different effects with small to moderate pairwise standardized mean differences for TRK, including ab front and back ext.

### 3.3. Running Economy (RE)

RE revealed a very large, though not statistically significant mode × time interaction effect. Pairwise post-hoc testing did not reveal any statistically different results (0.08 < *p* < 0.67). However, small to large standard mean differences as a measure of the between-mode effect sizes at the different time points were found for pair-wise comparison. Lastly, only small between-mode effect sizes were observed between UPR and TRK conditions. See Figure 1 for a summation of RE details. 

## 4. Discussion

The present study, a randomized controlled crossover pilot project, investigated the effect of 24-min upper body fatigue (UPR) and trunk fatigue (TRK) sessions on running economy (RE) in well-trained and competitive collegiate runners. We hypothesized both UPR and TRK would decrement RE vs. control (CON); however, our study indicated no significant differences between all conditions (though a large but not significant interaction occurred with regard to mode × time). Further, we surmised that TRK would significantly decrement RE vs. UPR and also exert a greater negative change (i.e., greater decrease in RFD from pre- to post-condition) on the core muscle group vs. UPR. This was partially supported. Despite no significant difference between UPR and TRK on RE, TRK significantly altered (and not UPR) ab flex and back ext max isometric RFD values from pre- to post-fatigue protocol. Notably, both fatigue conditions decremented RE (mL·min^−1^·kg^−1^) vs. CON with a small to large effect size across time points (i.e., 5, 10, and 15 min of RE at IAT). Hence, based on the aforementioned effect sizes, the fatigue protocols worked to effectively alter RE (for the worse) and in particular indicated a larger, specific overload/weakness in TRK, confirming the specificity of our fatigue sessions (i.e., TRK vs. UPR). 

RE is considered a major determinant of distance running success, accounting for as much as 30% of performance variation in elite level competitors [3]. Therefore, in order to optimize running potential through augmented RE, core (or TRK) strength and stability are often considered important contributors [17]. In fact, Sato and Mokha [17] found a 6-week core strength training program to significantly improve 5 km run time trial vs. control despite no improvement in kinematic measures (i.e., ground reaction forces). They did not evaluate TRK or core musculature involvement. To this end, core stability and core strength are routinely discussed in the literature or coaching circles as important to sport performance while lacking depth and breadth of research to confirm this claim [18]. We offer a pilot project whereby it seems core or trunk fatigue altered RE to a greater extent (though not significantly, but with small to large effect sizes, independent of sample size) than UPR or CON. Thus, because gross ml·kg^−1^·min^−1^ of oxygen uptake and use was greatest in TRK during the RE trial (especially vs. CON), oxygen delivery may have been altered or shunted to the highly fatigued, less efficient, and low economical core—thereby underscoring the need for a strong and stabilizing trunk to maintain run performance. We can only speculate that change in running mechanics and localized oxygen uptake (e.g., greater O_2_ uptake to the core muscles due to the fatigue protocol) occurred to lower RE. However, a group of researchers sought to “detect deviations in the dynamic center of mass (CoM) motion due to running-induced fatigue using tri-axial trunk accelerometry” [19]. This group found variability in horizontal plane trunk accelerations, with anteroposterior trunk accelerations to be less regular from step-to-step and not as predictable. They inferred that detectable alterations in CoM explained a fatigue state while running and that this could lead to biomechanical alterations in gait, thereby reducing running performance. Moreover, this could lead to increased energy expenditure (i.e, increased O_2_ cost) that is not beneficial for propulsion and thereby further encumber RE and performance outcomes.

Looking at the broad run performance picture, RE is affected by a myriad of variables, including genetics (affecting all subsequent parameters), metabolic efficiency, cardiorespiratory efficiency, training, biomechanical efficiency, and neuromuscular efficiency [3]. We believe, because both TRK and UPR body conditions trended toward worse RE vs. CON, with TRK eliciting the greatest drop in RE, that core/TRK muscle fatigue has the potential to collectively degrade all aforementioned RE factors (based on observed small to large effect sizes). The primary influence might be a shift in blood flow to the overly fatigued core (i.e., TRK condition), which may act as a lead domino, progressively toppling the aforementioned factors and leading to eventual inferior running performance vs. CON. This leads us to postulate that concerted, periodized core/TRK strength training, especially during the off- and pre-season as well as into a runner’s competitive season, has the potential to optimize RE by preventing a lead domino/fatigue influence. Sato and Mokha [17] agree, recommending up to a year of core strength training with episodic testing to monitor change in biomechanical parameters of running performance, which has the potential to improve running outcomes.

Lastly, in accordance with our RE assessment protocol, we used a high level performance pace (i.e., IAT) [15] to ensure valid, real world applicability of our results. To this end, because the TRK session effectively and significantly diminished core musculature isometric RFD from pre- to post-workout, each runner’s ability to maintain their IAT was effectively compounded. This further underscores the probable importance of maintaining a stable and strong core musculature, via specific and periodized strength-endurance training, for preserving race pace (e.g., in a 5 km or 10 km race).

## 5. Conclusions

The aim of this pilot study was to analyze the effect of TRK and UPR on RE. Our data suggested that running at IAT, a prominent endurance performance pace, seemed to be more strenuous due to reduced RE in TRK (vs. CON) based on small to large effect sizes across RE test time points. This may indicate a need to incorporate core/trunk strength training into a runner’s seasonal training routine to help optimize RE at IAT and, therefore, augment performance. Accordingly, adding some form of periodized, upper body strength-endurance training could also help offset negative RE influences (based on moderate effect sizes vs. CON). Future research should elucidate the mechanisms (e.g., metabolic efficiency, cardiorespiratory efficiency, neuromuscular efficiency, biomechanical efficiency) by which core strength (and/or upper body) training affects RE using a 6-week or longer, targeted routine in well-trained runners with periodic assessment of outcome measures (e.g., biomechanical analysis, running mechanics assessment). Lastly, a bigger sample size should help tease out statistically significant differences between conditions with, potentially, more pronounced effect sizes. 

## Figures and Tables

**Figure 1 sports-07-00195-f001:**
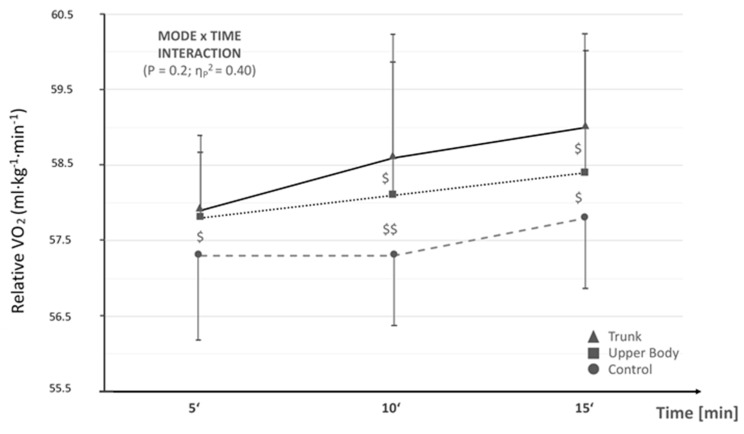
Relative running economy VO_2_ (mL/kg/min) at individual anaerobic threshold (IAT) velocity at 5, 10 and 15 min (i.e., VO_2_ averaged across 4–5, 9–10, and 14–15 min at IAT). Ṩ = small to moderate effect size; ṨṨ = large effect size. No statistically significant differences found between conditions.

**Table 1 sports-07-00195-t001:** Physiological characteristics of participants during incremental treadmill protocol to assess VO_2max_ and IAT.

Group	VO_2max_	RPE	HR_max_	VE/VO_2_	BR	IPE [La]
	ml·kg^−1^·min^−1^	Borg, 6–20	beats·min^−1^	no units	breaths·min^−1^	mmol·L^−1^
Males (n = 2)	67.4 (3.3)	20 (0.0)	189.0 (7.8)	26.3 (1.63)	52.8 (7.7)	14.0 (5.2)
Females (n = 5)	59.7 (5.4)	19.8 (0.5)	185.3 (6.1)	28.0 (4.12)	51.6 (7.0)	10.1 (2.4)
Combined (n = 7)	61.9 (5.9)	19.9 (0.4)	186.3 (6.12)	27.5 (3.5)	51.9 (6.5)	11.2 (3.4)

Date presented as mean ± (SD). IPE = immediate post-exercise. VO_2max_ = maximal oxygen uptake. RPE = rating of perceived exertion. HR_max_ = heart rate maximum. VE/VO_2_ = ventilatory equivalent of oxygen. BR = breathing rate. [La] = blood lactate accumulation.

**Table 2 sports-07-00195-t002:** Physiological observations during trunk and upper body fatigue protocols.

Condition	IPE RPE	IPE HR	IPE [La]
	Borg, 6–20	beats·min^−1^	mmol·L^−1^
TRK (n = 7)	17.4 (1.6)	128.0 (17.5)	2.9 (1.9)
UPR (n = 7)	19.0 (1.6)	114.0 (28.6)	3.5 (1.7)

Date presented as mean ± (SD). IPE = immediate post-exercise. RPE = rating of perceived exertion. HR = heart rate. [La] = blood lactate accumulation (note, resting values, prior to the fatigue protocol, were 1.1 mmol·L^−1^ for both conditions).

**Table 3 sports-07-00195-t003:** Maximal trunk strength for twist right, abdominal flex and back extension at pre- and post-testing during TRK and UPR conditions.

Trunk Exercise	TRK		UPR		*p*-Value	η_p_²
	**pre**	**post**	**pre**	**post**		
Twist, right [N]	1650 (380)	1556 (380)	1603 (350)	1598 (330)	**0.034**	**0.56**
Abdominal Flexion [N]	692 (182)	595 (133) *	673 (186)	646 (158)	0.064	**0.46**
Back extension [N]	1492 (319)	1252 (200) *	1404 (297)	1360 (204)	0.110	**0.44**

Data are presented as means with standard deviations (SD). Time × Condition interaction effects are presented as *p*-values and eta-square. Large effects and statistically significant effects are highlighted in bold. Post-hoc tests are indicated with *p* < 0.05 *.

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
