# Peer review of "Trunk and Upper Body Fatigue Adversely Affect Running Economy: A Three-Armed Randomized Controlled Crossover Pilot Trial"

_sports, 2019, doi:10.3390/sports7080195_

Round 1

Reviewer 1 Report

In the reviewer opinion, this paper addresses a critical issue in the determinants of performance in endurance athletes.

However, some elements of the methodological approach need to be clarified.

Material and Methods section

Study design and participants

- The plan was developed along one month. Did the authors control the training load during that períod? In endurance athletes during 4 weeks is possible to expect aerobic adaptations in submaximal intensities, that could affect the RE results. What did the authors do to control the training load impact?

- The authors mention that during the testing days participants wore an HR monitor (Lines  72-73). They do not present that data, and it is not clear what was the purpose. Can you please add more information? 

- Line 77, the authors mention that athletes are advised to restrain from intense exercise 24h before testing. However, the impact over some markers of muscle damage persists to 36-72hours, depending on the type of exercise. How the authors control this possible effect.

Additionally, females along the study period undergo in different phases of their menstrual cycle, which in some  athletes showed an impact over the performance capacity  

- Data that characterize males and females must be present separately. 

Testing procedures

- please add a reference for your  VO2max acceptance ( criteria)

- Blood samples were collected at the end of every step,  in the incremental protocol? Please clarify?

- Please provide information related to the  IAT determination

Control condition

- The authors report that in the control condition athletes only do some relaxing and stretching exercises. Are the authors aware of the eventual negative impact of stretching on power and strength activities?

Statistics 

- Please add a reference to the magnitude of mean differences (ES)

Results

Why the authors did not present data of the incremental protocol? Please consider presenting that data

Reviewer 2 Report

The study has sought to examine the influence of two exercise protocols of 24 min duration relative to a stretching control trial on running economy as a means to ascertain whether fatigue influences a key performance predictor. While the paper has some possible application to real-world training prescription, it is hindered by some serious flaws that limit generalisability.

A major consideration throughout the manuscript relates to the use of the term fatigue; the authors are actually assessing the influence of the two exercise protocols on the variables being examined. It is not actually clear whether the athletes are indeed fatigued given the absence of data supporting the validity of the “fatiguing” protocols used.

The rationale for the study is not developed fully and while a hypothesis and aim is presented, this is not derived from a comprehensive review of literature that leads to an understanding of why the study is needed. Notably, the main premise of the study is discussed very superficially and as mentioned in the abstract (L13), I question whether, based on the definition of running economy given, whether there is indeed a negative relationship with running economy and is it merely characteristics of running mechanics? As highlighted, running economy is the steady state oxygen consumption at a given submaximal running velocity, whereas I don’t think the practitioners are actually “seeing” this; perhaps they are observing altered running mechanics when key musculature is exposed to a prior exercise stimulus with incomplete recovery. While I understand that there may be few, if any, studies that have examined similar approaches, could the authors not derive data from previous work conducted outside of the immediate field? Likewise, the absence of data is not solely a rationale for a study – yet this seems to be the approach taken.

Akin to the comments above, the practical applications of this study are weak. The authors do not make clear a convincing argument that any transfer to practice is likely yielded from the findings.

The submission suffers from instances of poor English language communication. While authors whose first language is not English should not be discouraged from submitting to the journal, then there is a need to ensure that specific proof-reading services are employed to enhance readability of future submissions. For example, the opening lines of the discussion contain numerous spelling and grammatical errors.

Round 2

Reviewer 2 Report

My original suggestions have been satisfactorily addressed